# The Metallotolerance and Biosorption of As(V) and Cr(VI) by Black Fungi

**DOI:** 10.3390/jof10010047

**Published:** 2024-01-05

**Authors:** Cristy Medina-Armijo, Daniela Isola, Josep Illa, Anna Puerta, Marc Viñas, Francesc X. Prenafeta-Boldú

**Affiliations:** 1Program of Sustainability in Biosystems, Institute of Agrifood Research and Technology (IRTA), 08140 Caldes de Montbui, Spain; 2Faculty of Pharmacy and Food Sciences, University of Barcelona, 08028 Barcelona, Spain; 3Department of Economics, Engineering, Society and Business Organization (DEIM), University of Tuscia, 01100 Viterbo, Italy; 4Department of Computing and Industrial Engineering, University of Lleida, 25001 Lleida, Spain

**Keywords:** Chaetothyriales, *Exophiala mesophila*, fungal melanin, heavy metal bioremediation, hexavalent chromium tolerance, Langmuir biosorption isotherm, pentavalent arsenic tolerance, second-order kinetic model

## Abstract

A collection of 34 melanized fungi isolated previously from anthropogenic contaminated sites were assessed for their tolerance to toxic concentrations of As(V) and Cr(VI) anions. Three strains of the species *Cyphellophora olivacea*, *Rhinocladiella similis*, and *Exophiala mesophila* (Chaetothyriales) were identified as hyper-metallotolerant, with estimated IC_50_ values that ranged from 11.2 to 16.9 g L^−1^ for As(V) and from 2.0 to 3.4 g L^−1^ for Cr(VI). *E. mesophila* and *R. similis* were selected for subsequent assays on their biosorption capacity and kinetics under different pH values (4.0 and 6.5) and types of biomass (active and dead cells and melanin extracts). The fungal biosorption of As(V) was relatively ineffective, but significant removal of Cr(VI) was observed from liquid cultures. The Langmuir model with second-order kinetics showed maximum sorption capacities of 39.81 mg Cr^6+^ g^−1^ for *R. similis* and 95.26 mg Cr^6+^ g^−1^ for *E. mesophila* on a dry matter basis, respectively, while the kinetic constant for these two fungi was 1.32 × 10^−6^ and 1.39 × 10^−7^ g (mg Cr^6+^ min)^−1^. Similar experiments with melanin extracts of *E. mesophila* showed maximum sorption capacities of 544.84 mg Cr^6+^ g^−1^ and a kinetic constant of 1.67 × 10^−6^ g (mg Cr^6+^ min)^−1^. These results were compared to bibliographic data, suggesting that metallotolerance in black fungi might be the result of an outer cell-wall barrier to reduce the diffusion of toxic metals into the cytoplasm, as well as the inner cell wall biosorption of leaked metals by melanin.

## 1. Introduction

Heavy metals and metalloids (HMMs) are found naturally in the Earth’s crust at relatively diffused amounts, but they might become concentrated as a result of anthropogenic activities such as mining, agriculture, and husbandry, as well as several industrial manufacturing processes. Because of their inherent high toxicity, pollution with HMMs has become a pressing public health issue in many parts of the world [1]. Metal ions persist in the environment due to the bioaccumulation tendency of living organisms and their limited capacity to metabolize them into less toxic forms [2]. Once absorbed into the cell, HMMs bind to vital components, such as structural proteins, enzymes, and nucleic acids, impairing several fundamental metabolic functions [3].

Groundwater contaminated with arsenic is still poisoning millions of people, primarily from developing countries, and has been listed by the WHO as one of the ten chemicals of major public health concern [4,5]. Arsenic exposure to humans can be attributed to various sources, including natural deposits, agricultural pesticides, and industrial effluents [5]. Natural geochemical processes can also lead to toxic and subtoxic concentrations of arsenic in groundwater [6]. Arsenic-contaminated water typically contains both arsenate and arsenite species, As(V) and As(III). Water pollution with chromium is gaining more consideration because it is globally widespread and also represents an important public health concern [7,8]. The majority of chromium released into the environment originates from industrial activities, particularly from the processing and manufacturing of chemicals, minerals, steel, leather tanning, textile dyeing, metallurgical operations, and various other industrial processes [7]. The most stable oxidation states of chromium in water correspond to the hexavalent Cr(VI) and trivalent Cr(III) species, but concerns are primarily related to Cr(VI) owing to its high biotoxicity.

A number of technical approaches for the remediation of HMM pollution and water purification have been proposed [9]. These solutions should rely on technically and economically viable methods, such as the biosorption of HMMs into readily available biomass sources. Many reports in the literature describe the capacity of pure cultures of bacteria [10,11], algae [12], aquatic plants [13], and fungi [14,15] to remove HMM ions from aqueous solutions. Such bioremediation alternatives are an interesting option for the decontamination of water and soil because these processes require few reagents and energy, they generate relatively low amounts of toxic waste products, and because biosorption is highly effective in reducing HMMs at relatively low concentrations [1,13]. However, the biotechnological removal of HMMs still faces some scalability problems associated with the difficulty to find suitable microorganisms that are able to cope with a highly oligotrophic and toxic environment while having a high HMM biosorption potential.

In this context, one particular group of ascomycetes known as black fungi (BF), because of their common polymorphism as yeast-like, hyphal, and meristematic growth, have been identified recently for their capacity to bind heavy metals to their cell wall [16,17]. These fungi are characterized by a high morphological and metabolic plasticity and poly-extremotolerant traits, which allow them to colonize a diverse range of often divergent and uncommon habitats [18]. The main defining character of BF is a strongly melanized cell wall, which confers them protection against extreme environmental conditions, such as exposure to UV and ionizing radiation, desiccation, cold/hot temperatures, high salinity, and oligotrophic environments [19,20]. This latter trait has also been linked to the ability to use toxic volatile hydrocarbons as the only source of carbon and energy [21,22]. As for their identities, BF primarily fall into two main phylogenetic groups: the Eurotiomycetes (Chaetothyriales) and the Dothideomycetes (Dothideales, Cladosporiales, and a few other related orders in the Mycophaerellales) [18,23].

The structure of fungal melanin is somewhat similar to soil humic acids with respect to volatile compounds released upon pyrolysis and amino acid hydrolysis [24]. In particular, the melanin of BF is composed of short-distance non-hydrolysable strong carbon–carbon bonds based on 1,8-dihydroxynaphthalene (DHN), modified with different functional groups, such as carboxyl, phenolic, hydroxyl, and amino [25,26], which provide many potential binding or biosorption sites for metal ions [27]. Special attention has been given to two main functions attributed to melanin in relation to metal ions: as a reservoir for the temporary storage and release of certain nutrients, and as chelating agents of HMMs for protecting the cell against metal toxicity. However, the biodiversity of metallotolerant BF and their HMM biosorption potential have seldom been investigated.

In this study, we screened a collection of BF species isolated previously from diverse anthropogenic-polluted sites for their tolerance to As(V) and Cr(VI), selected, respectively, as a model heavy metal and metalloid. The biosorption potential was determined on selected metallotolerant strains, using living fungal cultures, dead biomass, and melanin extracts. The obtained results could give new insights into the biology of BF and might contribute to the development of biotechnological applications. 

## 2. Materials and Methods

### 2.1. Biological Material

A culture collection of 34 BF strains confidently identified at the species level were used (Table 1). Most of these strains were isolated during previous studies [21,28,29] from car fuel dispensers and tanks, stone buildings exposed to pollution and toxic biocides, and washing machines. These strains are currently maintained at the Culture Collection of Fungi from Extreme Environments (CCFEE) at the Tuscia University in Viterbo, the Westerdijk Fungal Biodiversity Center (formerly Centralbureau voor Schimmelcultures—CBS), and the Institute of Agrifood Science and Technology (IRTA).

### 2.2. Metallotolerance Assays

The capacity of the collected fungal strains to grow in the presence of increasing concentrations of As(V) and Cr(VI) was evaluated on solid cultures. Into Petri dishes (12 cm in diameter), we poured potato dextrose agar (PDA; Condalab, Torrejón de Ardoz, Spain) supplemented with 2.5, 5.0, 7.5, 10.0, or 12.5 g of As^5+^ L^−1^ from sodium arsenate (Na_2_HAsO_4_·7H_2_O; Thermo Fisher Scientific, Kandel, Germany) and 0.1, 0.5, 1.0, 1.5, and 2.5 g of Cr^6+^ L^−1^ from potassium dichromate (K_2_Cr_2_O_7_; Scharlab ExpertQ^®^, Sentmenat, Spain). Each BF strain was inoculated six times with an inoculation loop on each agar plate containing a defined concentration of As(V) or Cr(VI). The radial growth of each fungal colony was measured by averaging orthogonal diameters, and this value was then averaged again for all six colonies from every single plate. These measurements were repeated after 7, 15, 30, 45, and 60 days of incubation, which was performed at 25 °C under dark conditions. This prolonged timeframe was established to account for the potential long-term adaptation of slow-growing BF strains. Unamended control plates were also included. The tolerance index (TI) was calculated for every strain by dividing the measured growth when exposed to the metal in relation to the control plates. 

### 2.3. Production of Fungal Biomass

Fungal biomass for biosorption experiments on selected BF strains was produced in 0.5 L batch liquid cultures incubated at room temperature under shaking conditions (80 rpm) for 10 days. Yeast extract (4 g L^−1^) was supplied as the carbon and energy source, and macronutrients were added in the form of 4.5 g of KH_2_PO_4_, 0.5 g of K_2_HPO_4_, 2.0 g of NH_4_Cl, and 0.1 mg of MgSO_4_·7H_2_O per liter. Mineral micronutrients were added in the form of 2 mL of a stock solution that contained 120 mg of FeCl_3_, 50 mg of H_3_BO_3_, 10 mg of CuSO_4_·5H_2_O, 10 mg of KI, 45 mg of MnSO_4_·H_2_O, 20 mg of Na_2_MoO_4_·H_2_O, 75 mg of ZnSO_4_·H_2_O, 50 mg of CoCl_2_·6H_2_O, 20 mg of AlK(SO_4_)_2_·12H_2_O, 13.25 g of CaCl_2_·H_2_O, and 10 g of NaCl per liter. Spore suspensions of pure cultures (0.5 mL, >10^6^ CFU mL^−1^) were used as inoculant. After 10 days of incubation, the fungal biomass was harvested by centrifugation (20 min at 4000 rpm) and washed with milliQ sterilized water. This process was repeated three times before using fungal biomass in batch biosorption assays.

### 2.4. Extraction and Purification of Melanin

Melanin was extracted from pre-grown BF cultures by adapting an acid hydrolysis method used previously with other fungi [30,31]. Briefly, harvested fungal biomass was homogenized in 100 mL of 1 M NaOH (120 rpm for 10 min) and treated with hot alkali (1 M NaOH at 121 °C for 20 min). The resulting suspension was centrifuged at 10,000 rpm for 10 min to remove fungal biomass, and the brownish liquid fraction was acidified to pH 2.5 with HCl 6 N and incubated for 12 h at 100 °C. The resulting black precipitate was centrifuged (4000 rpm for 20 min) and washed with deionized water three times. The precipitate was then lyophilized at a pressure of 0.7 mBar at −50 °C for 24 h, and the obtained melanin powder was kept at −20 °C until use.

### 2.5. Biosorption Assays

The capacity of the fungal biomass to accumulate As(V) and Cr(VI) ions was tested on a oligotrophic liquid medium to minimize growth. Consequently, the number of sorption sites remained consistent. This medium contained a buffer of 35 mM K_2_HPO_4_/NaH_2_PO_4_·2H_2_O (pH 6.5), glucose (0.3%), and yeast extract (0.01%), along with 20 to 200 mg L^−1^ of As(V) or Cr(VI), added as Na_2_HAsO_4·_7H_2_O or K_2_Cr_2_O_7_, respectively. Biomass from living and heat-inactivated fungal cultures (on a dry matter basis, approx. 50 mg DM) was resuspended into the liquid mineral medium and incubated in serum flasks (30 mL) under sterile conditions on a shaker (80 rpm at 25 °C). Experiments were carried out in triplicate, and liquid samples were taken after 2, 5, 7, 14, 21, and 30 days of incubation to measure the concentration of As(V) and Cr(VI). Biosorption experiments were repeated at pH 4.0 by adding 0.1 M HCl.

The assays for the biosorption of Cr(VI) onto melanin extracts were performed with 0.3 g of melanin powder resuspended in 50 mL of a solution (pH 6.5; 150 rpm; 25 °C). The initial solution’s Cr(VI) concentration was 30 mg Cr^6+^ L^−1^ and incubations lasted up to 72 h. Incubations were carried out in triplicate, and liquid samples were taken regularly to measure the time-course evolution of the concentration of Cr(VI).

### 2.6. Analytical Methods

The content of total arsenic in liquid culture supernatant was determined using Flame Atomic Absorption Spectroscopy (Model SpectrAA-110, Varian, Mulgrave, Australia). Chromium (VI) in the liquid fraction was determined with a colorimetric method based on the reaction with the complexing agent 1,5-diphenylcabazide (Sigma-Aldrich, St. Louis, MI, USA) that forms a purple-violet-colored complex, which was quantified by measuring the absorbance at a wavelength of 540 nm using a spectrophotometer (model EMC-11S UV brand, Duisburg, Germany).

### 2.7. Numerical Methods and Statistical Analysis

The software GraphPad Prism version 8.0 (GraphPad Software, San Diego, CA, USA) was used for performing one-way ANOVA (multiple comparisons Tukey test) and for the calculation of the metal concentration causing a 50% inhibition of fungal growth (*IC*_50_). After a given time of incubation, *IC*_50_ was calculated by fitting the tolerance index (*TI*) measurements at different HMM concentrations (*C_t_*) to a two-parameter Hill logistic model (*IC*_50_ and slope factor *h*; Equation (1)).
(1)$$TI=\frac{1}{{\left(1+\frac{{IC}_{50}}{C_{t}}\right)}^{h}}$$

As for the biosorption assays, the specific amount of the HMMs that is taken up at any given time (*q_t_*) by a defined amount of fungal biomass *M_a_*, when incubated at an initial concentration *C*_0_ in batch incubations of liquid volume *V*, is a function of the remaining HMM concentration (*C_t_*) as the experimentally measured variable, as described by Equation (2).
(2)$$q_{t}=\frac{V\cdot (C_{0}-C_{t})}{M_{a}}$$

The sorption rate is usually described either as first- or second-order kinetics (Equation (3)), where *k_i_* is the kinetic constant at the considered *i*th order, and *q_e_* is the adsorbed metal fraction when reaching equilibrium with the concentration *C_e_* of metal in solution. The integration of Equation (3), under the hypothesis of constant *C_e_*, by considering second-order kinetics gives Equation (4), which relates *q_t_* with time. Equation (4) is used to produce a linear plot (*t*·*q_t_*^−1^) versus *t* with experimental data that allows the calculation of the two parameters *k*_2_ and *q_e_* from the intercept and slope.
(3)$$\frac{dq_{t}}{dt}=k_{i}\cdot {\left(q_{e}-q_{t}\right)}^{i}$$
(4)$$\frac{t}{q_{t}}=\frac{1}{k_{2}\cdot q_{e}^{2}}+\frac{t}{q_{e}}$$

When the concentration of metal in solution varies with time, the adsorbed fraction at equilibrium *q_e_* is usually modelled as a function of liquid equilibrium concentration *C_e_* by the Langmuir or Freundlich isotherm (Equations (5) and (6)). Both functions have two unknown parameters, *q*_max_ and *K_L_* in the Langmuir equation and *K*_F_ and *n* in the Freundlich equation.
(5)$$q_{e}=q_{max}\frac{K_{L}\cdot C_{e}}{{1+K}_{L}\cdot C_{e}}$$
(6)qe=KL·Ce1n

The substitution in Equation (3) of *q_e_* by Equations (5) or (6), the Langmuir or Freundlich isotherms named here as *q_e_*(*C_t_*), and *q_t_* by Equation (2) results in the differential Equation (7). This new expression describes the temporal evolution of the concentration in the liquid phase depending on the assumed sorption isotherm model and kinetic order.
(7)$$\frac{dC_{t}}{dt}=-k_{i}\cdot {\left(\frac{M_{a}}{V}\cdot q_{e}\left(C_{t}\right)\cdot \left(C_{0}-C_{t}\right)\right)}^{i}$$

For the given set of 3 parameters, Equation (7) was numerically integrated along time and the goodness of the fit to the measured *C_t_* values evaluated as the sum of square errors. A Matlab (version R2017a) routine was used for this purpose to find the parameter values that produced the best fit.

## 3. Results and Discussion

### 3.1. The Metallotolerance of Black Fungi

Of the 34 studied BF (Table 1), 28 strains were able to grow on some of the tested concentrations of As(V) and/or Cr(VI) (Table 2). Seventeen strains (50%) displayed measurable growth on agar cultures exposed to As(V) at the maximum tested concentration of 12.5 g of As^5+^ L^−1^, while eight strains (23.5%) showed no growth under the minimum tested concentration of 2.5 g of As^5+^ L^−1^ (Table 2). The first group of As(V)-tolerant strains primarily belonged to the Chaetothyriales (*Cyphellophora olivacea*, *Exophiala crusticola*, *E. equina*, *E. lecanii-corni*, *E. mesophila*, *E. oligosperma*, *E. phaeomuriformis*, *E. xenobiotica*, and *Rhinocladiella similis*), though some fungi from the Dothidiomycetes were found to be metallotolerant as well (*Aulographina pinorum*, *Aureobasidium melanogenum*, *Coniosporium uncinatum*, and *Neohortaea acidophila*). A few chaetotyrialean species were also found to be slightly As(V) tolerant (*E. angulospora*, *E. heteromorpha*, and *Knufia epidermis*), but representatives of the order Dothidiomycetes were predominant in this group (*Cladosporium herbarum*, *Scolecobasidium globalis*, and *Rhizosphaera kalkhoffii*). As for Cr(VI), only 5 chaetothyrialean strains (*Rhinocladiella similis*, *Cyphellophora olivacea*, *Exophiala mesophila*, *E. crusticola,* and *E. lecanii-corni*) were able to grow at the highest tested concentration of 2.5 g of Cr^6+^ L^−1^, while 12 strains showed measurable growth at the lowest tested concentrations of 0.1 g of Cr^6+^ L^−1^ (*Au. pullulans*, *C. uncinatum,* and *Cl. herbarum* in the Dothidiomyces; *K. epidermis*, *E. equina,* and *E. phaeomuriformis* in the Chaetothyriales).

The tolerance index (TI) of fungi to the tested concentrations of As(V) and Cr(VI) and the estimated 50% inhibitory concentration of these compounds (IC_50_) after 30 days of incubation are summarized in Table 2. The 30-day incubated period was taken as an intermediate reference between short-term and long-term toxicity. When considering all incubation times, most fungi showed a negative correlation between TI values and exposure time to high HMM concentrations (Appendix A), which indicates that HMM toxicity tends to manifest after prolonged incubations. A few chaetothyrialean strains stood out from the rest because of their remarkable level of metallotolerance, such as *Exophiala crusticola* CCFEE 6188. This particular strain had the highest tolerance to As(V), with average TI values that after 30 days ranged from 1.00 at 2.5 g of As^5+^ L^−1^ to 0.50 at 12.5 g of As^5+^ L^−1^. The corresponding modelled IC_50_ estimate was 10.0 g of As^5+^ L^−1^. The number of available strains of *E. crusticola* is relatively small, so little information is available on the ecophysiology of the species. The type strain was isolated from a biological soil crust sample of the Colorado Plateau and Great Basin desert [32]. More recently, this species has also been reported in the Atacama Desert, near Calama in Chile [33], which is among the driest sites in the world and lies close to a large open-pit copper mine. This environment illustrates the polyextremophilic nature of chaetothyrialean fungi to withstand drought, UV radiation, and exposure to toxic metals. 

Conversely, the tolerance of *E. crusticola* CCFEE 6188 to Cr(VI) was lower when compared to other fungi, as it had a TI of 0.73 at 0.1 g of Cr^6+^ L^−1^ after 30 days of exposure but it grew scarcely at the higher tested concentrations (its IC_50_ was 0.56 g of Cr^6+^ L^−1^). There were three other strains that displayed a remarkable tolerance to both As(V) and Cr(VI), showing limited inhibition with respect to the control (Figure 1): *R. similis* CCFEE 6361, *C. olivacea* CCFEE 6619, and *E. mesophila* IRTA M2-F10. Their TI values were above 0.4 at the tested concentrations of As(V) and Cr(VI), and the IC_50_ was in the range of 6–48 g of As^5+^ L^−1^, though the latest estimate must be taken with caution as it is well above the highest tested concentration, 0.05–0.25 g of Cr^6+^ L^−1^ after 30 days of exposure (Table 2). These strains were generally characterized by TI values at the highest As(V) and Cr(VI) tested concentrations that tended to increase with longer incubation times, up to 60 days (Appendix A). This phenomenon indicates a progressive longer-term adaptation to HMMs of these particular strains, in contrast to most of the tested BF collection.

Literature data on the fungal toxicity of HMMs are scarce, particularly with melanized fungi despite their polyextremophilic nature and association to toxic chemicals [21]. There are some reports on the fungal toxicity of arsenic oxyanions, but those have often been performed at a rather low milligram per liter range (between 10 and 500 mg L^−1^) [34,35,36,37,38]. A few surveys were carried out at the gram per liter range, however. A screening of fungi isolated from arsenic-polluted soil for As(V) tolerance resulted in the selection of five strains, dubbed as “hyper-tolerant”, that had TI values at 10 g of As^5+^ L^−1^ that ranged from 0.19 to 0.31 for *Aspergillus* sp., *Neocosmospora* sp., *Rhizopus* sp., and *Penicillium* sp., and up to 0.96 for an unidentified sterile fungus [15]. Singh et al. [39] reported nine fungal strains that were tolerant to As(V) up to 10 g of As^5+^ L^−1^, which belonged to the genera *Trichoderma*, *Aspergillus*, *Rhizopus*, *Microdochium*, *Chaetomium*, *Myrothecium*, *Stachybotrys*, *Rhizomucor*, and *Fusarium*. However, no quantitative tolerance parameters were derived from this study.

Chromium has a complex valence layer that produces different oxidation states that interact with specific nutrients, accentuating the toxicity of this metal [3]. Several previous reports have corroborated the severity of Cr(VI) toxicity to fungi, when compared to As(V). In general, Cr(VI) is more toxic than As(V) to fungi because of its higher reactivity and capacity to generate oxygen reactive species (ROS), which can disrupt several metabolic functions [3,8]. For example, out of 14 isolates from tannery effluents contaminated with Cr(VI), only 1 strain of *Trichoderma viride* (fam. Hypocreaceae) was able to show some growth at 1 g of Cr^6+^ L^−1^ when cultured under laboratory conditions [40]. Other species in this genus have been evaluated for Cr(VI) tolerance, such as a *T. harzianum* strain isolated from an HMM-polluted mine [41], which displayed a TI at 1 g of Cr^6+^ L^−1^ as low as 0.024. Other *Aspergillus* spp. were also tested in that study (*A. sclerotiorum*, *A. aculeatus*, and *A. niger*) and yielded higher TI values, between 0.12 and 0.67, depending on the strain and the tested Cr(VI) concentration. Another strain of *T. viride* isolated from tannery wastewaters displayed TI values of 1.15, 0.13, and 0.08, as determined from the biomass production from liquid cultures, after 21 days of incubation with 50, 500, and 1000 mg of Cr^6+^ L^−1^ [40]. A second Cr(VI)-tolerant strain identified as *Penicillium citrinum* showed a similar profile. Interestingly, those fungi seemed to be biostimulated at low concentrations of Cr(VI), both in terms of biomass production and secreted laccase enzymes.

None of the previously mentioned taxa are BF and, in fact, there are few reports quantifying the metallotolerance within this particular group of fungi. The chaetothyrialean *Exophiala pisciphila* has been isolated repeatedly from the roots of plants growing on soils that are polluted with heavy metals, and in vitro analyses have shown that this fungus tolerates concentrations of Pb(II), Cd(II), and Zn(II) at an IC_50_ of 0.8, 0.3, and 1.5 g L^−1^, respectively [42]. Concerning the tolerance to As(V), one study with liquid cultures of different strains of *E. sideris* isolated from HMM-polluted environments reported IC_50_ values between 2.0 and 3.7 g of As^5+^ L^−1^, depending on the isolate [43]. These latter results are in the As(V) IC_50_ range found in our study for the strains that belong to the *Exophiala* genus (Table 2). As for specific accounts on the tolerance of BF to Cr(VI), a minimum inhibitory Cr(VI) concentration of 300 mg of Cr^6+^ L^−1^ was determined for the growth of a strain identified as *Cladosporium perangustum* (fam. Cladosporiaceae) [44].

Some of the BF included in this study encompassed multiple strains from the same species (i.e., two strains of *Au. pullulans*, *C. uncinatum*, *A. pinorum*, *E. heteromorpha*, *E. mesophila*, and *E. oligosperma*, and four of *E. xenobiotica* and *K. epidermis*). Comparing growth inhibition patterns among these strains revealed a wide intraspecific variability in their tolerance to HMMs (Table 2). Examples of the most disparate cases include *Au. pullulans* and *C. uncinatum*, with one of the two tested strains (CCFEE 6244 and CCFEE 5820) of each species able to grow in the presence of As(V) and Cr(VI), while the other two (CCFEE 5876 and CCFEE 6149) did not show any growth at all. The “hyper-tolerant” *E. mesophila* strain IRTA M2-F10 also differed significantly from the conspecific CCFEE 5690. 

This observation deserves further investigation to verify whether the recorded variability in metal tolerance is intrinsic to the species considered or if it is due to the lack of knowledge in the identification of related species. On the one hand, a detailed molecular analysis could be useful in better determining the position occupied by the two strains within the large group of *Au. pullulans*. On the other hand, it could be valuable in defining the close relatives of *C. uncinatum*, which are currently unknown. 

On the contrary, metallotolerance appears to be relatively conserved in the four tested strains of *E. xenobiotica*, which were all able to grow at 12.5 g of As^5+^ L^−1^ and displayed IC_50_ values of 0.7–3 g of As^5+^ L^−1^, but none of them grew in the presence of Cr(VI). The two available strains of *E. oligosperma* displayed a somewhat similar growth pattern in the presence of As(V), with IC_50_ values of 0.2–2.5 g of As^5+^ L^−1^, but neither grew on any of the tested concentrations of Cr(VI). These strains are very similar to each other when comparing their ITS sequences. 

At the intragenus level, significant variability in HMM tolerance is also manifested in *Exophiala,* which, along the observed diverse degrees of metallotolerance, also includes species that, like *E. heteromorpha,* were consistently unable to grow at any tested concentration of As(V) and Cr(VI). The observed intra- and interspecific differences in HMM tolerance might be attributed to diverse degrees of adaptation to HMM-polluted environments, due to specific genetic changes, epigenetic regulation, and phenotypic adaptations to stressful conditions [30]. Melanization is often cited as a feature that confers tolerance to HMMs because of its capacity to absorb toxic metals [28,45]. However, all tested strains were conspicuously melanized and were isolated from similar environments exposed to toxic chemicals. Hence, besides melanin production and adaptation to toxic environments, other factors must contribute to the tolerance of HMMs in BF. Earlier studies have proposed an array of multiple mechanisms that enable fungi to cope with HMMs [3,46,47], such as reducing the basal energy for metabolism, activating protein protection and DNA repair against oxidative stress, enhancing iron and sulfur acquisition, transforming metal species to less toxic or volatile metabolites, detoxifying free radicals, and through homeostasis.

Exposure to As(V) and Cr(VI) caused macroscopic morphological changes in fungal growth that were visible on agar colonies, as shown in Figure 2 for the metallotolerant *R. similis* CCFEE 6361, *C. olivacea* CCFEE 6619, and *E. mesophila* IRTA M2-F10. Fungal biomass tended to display a stronger dark pigmentation upon metal exposure, which might be attributable to an increased level of melanin biosynthesis as a defensive mechanism. A few strains like those of *E. mesophila* also formed colonies that were irregular in shape and displayed coarser edges under stressful conditions, a phenomenon known as meristematic growth. This morphological plasticity has also been observed with *E. oligosperma* CCFEE 6327 when grown at 35 °C under laboratory conditions, a temperature close to its upper temperature growth limit [48]. A similar morphology is also manifested in the case of opportunistic mammal infections, defined then as muriform cells [49]. The isodiametric growth and the aggregated, compact shape of fungal microcolonies ensure the optimal surface/volume ratio, minimizing the direct exposure to external stressors [50]. This type of growth is also expressed as an adaptation in the closely related lithobiontic black fungi (known as rock-inhabiting fungi, RIF) [51].

### 3.2. Biosorption Assays

Two of the strains screened previously for metallotolerance, *R. similis* CCFEE 6361 and *E. mesophila* IRTA M2-F10, were selected for subsequent biosorption experiments because of their intrinsic tolerance to both As(V) and Cr(VI) and for their easy cultivation in liquid media for producing biomass. After 30 days of incubation of pre-grown fungal liquid cultures with As(V) and Cr(VI) (approx. 50 mg of DM L^−1^; an initial HMM concentration of 20 mg L^−1^), the HMM content remaining in the supernatant was measured (Table 3). From these results, it was evident that BF biomass had a comparatively low absorption capacity for As(V) when compared to Cr(VI), as removal efficiencies for the first did not exceed 10%, while for the second, they were higher than 80% under similar test conditions.

The specific As(V) removal capacity of living cultures of *E. mesophila* and *R. similis* incubated for 30 days at pH 6.5 was 1.07 and 1.34 mg of As^5+^ g DM^−1^. The difference between the two fungi was not statistically significant (*p* > 0.05). However, the biosorption of As(V) by heat-inactivated biomass of *E. mesophila* was significantly lower than that by living cultures of the same fungus. Despite the apparently modest As(V) biosorption results with the tested BF, previous similar studies with this metalloid have yielded even lower numbers. Different fungal species in the genera *Neocosmospora*, *Sordaria*, *Rhizopus*, and *Penicillium* displayed biosorption capacities that ranged from 0.009 to 0.016 mg As^5+^ g DM^−1^ [15]. Other authors claimed that cultures of fungi belonging to *Aspergillus*, *Fusarium*, *Rhizomucor,* and *Emericella* were able to absorb between 0.023 and 0.259 mg As^5+^ g DM^−1^ depending on the strain [39]. Those experiments were performed within the pH range of 4–7 used in our study, and all used strains correspond to fungi that are not conspicuously melanized. Hence, fungal melanization might indeed improve As(V) biosorption by cultures of BF.

Fungal biosorption patterns changed completely when Cr(VI) was used. Equivalent incubations with this metal showed that the specific removal capacity after 30 days of incubation was slightly higher for *E. mesophila* (12.57 mg Cr^6+^ g DM^−1^) than for *R. similis* (11.50 g Cr^6+^ mg DM^−1^). As with As(V), the heat inactivation of cultures of *E. mesophila* caused a reduction in the biosorption capacity (9.59 g Cr^6+^ mg DM^−1^). However, differences in the specific Cr(VI) biosorption capacity in all tested fungi and incubation conditions were statistically not significant (*p* > 0.05). Previous reports on the specific Cr(VI) biosorption capacity of fungi correspond primarily to modified biomass, and relatively few records with living fungal cultures are available. Some fungi isolated from samples of sludge and industrial effluents contaminated with heavy metals (*Trichoderma viride*, *T. longibrachiatum*, *Aspergillus niger*, and *Phanerochaete chrysosporium*) displayed biosorption capacities that ranged from 0.03 to 0.55 mg Cr^6+^ g DM^−1^ when incubated (150 rpm at 28 °C) with potato dextrose broth containing 50 mg Cr^6+^ L^−1^ for 4 days [52]. Lotlikar et al. [53] isolated three strains from Arabian Sea sediments, identified as *Purpureocillium lilacinum*, *Aspergillus sydowii,* and *A. terreus*, that were able to grow with 300 mg Cr^6+^ L^−1^ (pH 5, shaken conditions, and room temperature) and biosorbed 8, 10, and 13 mg Cr^6+^ g DM^−1^, respectively, after 20 days of incubation. Reports on Cr(VI) biosorption by BF are very limited; liquid cultures of *Aureobasidium pullulans* growing on the acid hydrolysate of peat containing HMMs (200 rpm; 26 °C; pH 6.0) were able to absorb 0.77 mg Cr^6+^ g DM^−1^ after 160 h of incubation [54]. 

The sorption of As(V) and Cr(VI) onto organic materials has been described as a pH-dependent phenomenon [55,56]. At acidic conditions, As(V) exists primarily in the form of dihydrogen arsenate (H_2_AsO_4_^−^), while Cr(VI) is present as chromate (CrO_4_^2−^). As the pH increases, H_2_AsO_4_^−^ transforms into hydrogen arsenate (HAsO_4_^2−^) or arsenate (AsO_4_^3−^) and CrO_4_^2−^ transforms into dichromate (Cr_2_O_7_^2−^), resulting in an increase in the net negative charge. The pH also influences the surface charge of the sorbent, so the point of zero charge (pzc)—the pH at which the net charge of the total absorbent’s surface is equal to zero—for fungal biomass is in the range of pH 4.0–4.5 [57,58]. Above these pH values, the surface becomes negatively charged, exacerbating its electrostatic repulsion towards As(V) and Cr(VI) anions. Interestingly, fungal incubations at pH 4.0 significantly (*p* < 0.05) enhanced the biosorption of Cr(VI), but not that of As(V), compared to analogous experiments at pH 6.5 (Table 3).

The HMM biosorption capacity of heat-inactivated biomass of *E. mesophila* was lower than with equivalent viable cultures (Table 3). Several investigations on the passive adsorption process on the cell surface have been performed using dead fungal biomass [59,60,61]. It has been claimed that the biosorption of dried biomass increases the contact surface between HMMs and the metal binding sites of the fungal cell wall [62]. However, heat-inactivated biomass may still keep the structures of the fungal cell relatively intact. A comparison between living cultures and dried pulverized biomass of a group of fungi (*Aspergillus foetidus*, *A. niger*, *A. terricola*, *Acremonium strictum*, *Paecilomyces variotii*, *Phanerochaete chrysosporium*, *Aureobasidium pullulans*, and *Cladosporium resinae*) claimed that while the Cr(VI) biosorption capacity in the first ranged from 0.1 to 3.0 mg Cr^6+^ g^−1^-DM, in the second, it increased up to 11.2 mg Cr^6+^ g^−1^-DM (pH 3; 100 rpm; 28.1 °C) [63]. Literature accounts on the use of inactivated fungal biomass as HMM sorbents have often included the addition of specific coadjuvants that mitigate the electrostatic repulsion between the sorbent and sorbate. A *Penicillium chrysogenum* was modified with three different surfactants (amines) to increase the removal capacities of modified biomass from 37.85 to 56.07 mg As^5+^ g^−1^ at pH 3 [64].

### 3.3. Biosorption Isotherms and Kinetics

The differential Equation (7) describing the first- and second-order kinetics, integrating the Langmuir and Freundlich isotherms, was fitted to experimental data on the time-course evolution of metal concentration in the supernatant. Changes with As(V) were too subtle to fit any sorption model with enough confidence, but the experimental data on the time-course depletion of Cr(VI) showed that the best fits were obtained with second-order sorption kinetics and the Langmuir sorption isotherm (Figure 3). Previous similar studies that compared different models to describe the fungal biosorption of metals showed that the Langmuir isotherm and second-order sorption kinetics displayed the best coefficients of determination [65,66,67]. The Langmuir isotherm is a theoretical approximation that considers a finite monolayer of available adsorption sites onto a homogeneous surface. This assumption implies that the sorbent has a limited surface-dependent maximum theoretical sorption capacity (*q*_max_), and that there is neither interaction nor transmigration of metal ions after monolayer adsorption.

The fitted parameters of Langmuir isotherms and second-order kinetics for the biosorption of Cr(VI) with the tested BF are summarized and compared with bibliographic data in Table 4. The “theoretical” maximum sorption capacity *q*_max_ of *R. similis* and *E. mesophila* (39.81 and 95.26 mg Cr^+6^ g DM^−1^) is above those seen previously with several other non-melanized fungi in analogous experiments. Most of those assays were performed with inactivated/modified fungal biomass and at a low pH, conditions that might favor biosorption, but under which process scaling-up into practical applications is challenging. In what concerns the half-saturation concentration (the metal concentration in equilibrium at which the biosorbed metal equals half of *q*_max_), which is the reverse of the affinity constant *K*_L_, both tested BF displayed rather similar values of 8.90 and 5.09 mg Cr^6+^ L^−1^, respectively, for *R. similis* and *E. mesophila*. The affinity to Cr(VI) of biomass from hyaline species tested previously is quite low in general when compared to BF (Table 4).

The second-order kinetic constant *k*_2_, which corresponds to the specific biosorption rate, is one order of magnitude lower for *E. mesophila* when compared to *R. similis*, 1.39 × 10^−7^ versus 1.32 × 10^−6^ g DM (mg Cr^6+^ min)^−1^. The higher *q*_max_ and lower *k*_2_ of *E. mesophila* when compared to *R. similis* could be related to the higher Cr(VI) tolerance of the former over the latter (Table 2). Several previous accounts on the second-order kinetics of Cr(VI) biosorption by non-melanogenic fungi have reported *k*_2_ values above 10^−2^ g DM (mg Cr^6+^ min)^−1^ (Table 4). Interestingly, previous studies with melanized fungal structures, like lyophilized cells of the BF *Cladosporium cladosporioides* and spores of *Aspergillus niger*, also yielded comparatively low biosorption rates and high substrate affinity.

In order to gain a deeper insight into the role of fungal melanin when exposed to Cr(VI), biosorption experiments were repeated with melanin extracts from *E. mesophila* (Figure 4). The obtained fitted parameter values are between 1 and 2 orders of magnitude higher than those measured with living cultures of the same fungus (*q*_max_ = 544.84 mg Cr^6+^ g DM^−1^; *K_L_* = 0.0075 L mg^−1^; and *k*_2_ = 1.67 × 10^−6^ g DM (mg Cr^6+^ min)^−1^). Considering that the mass of the extracted melanin corresponded to 12.5% of the fungal biomass, on a dry matter basis (125 mg g DM^−1^), the potential contribution of melanin to the biosorption capacity of whole cells could be as high as 71.5%. These results confirm that fungal melanin plays a vital role in the biosorption of HMMs. Furthermore, the *q*_max_ of melanin is well above that of several tested Cr(VI) organic absorbents, from raw and modified lignocellulosic materials [68] to advanced carbon nanomaterials [69].

**Table 4 jof-10-00047-t004:** Fitted parameters for modelling the biosorption of Cr(VI) using the second-order kinetic model and/or the Langmuir isotherm by different fungal species.

Fungus ^a^	pH	C (mg L^−1^)	Fitted Equation	Model Parameters				Source
				*k*_2_ (g (mg min)^−1^)	*q*_max_ (mg g^−1^)	*K*_L_ (L mg^−1^)	*r* ^2^	
*Rhinocladiella similis* (CS)	6.5	25	Equation (7)	1.32 × 10^−6^	39.81	0.1124	0.977	This study
*Exophiala mesophila* (CS)	6.5	25	Equation (7)	1.39 × 10^−7^	95.26	0.1964	0.924	This study
*Exophiala mesophila* (MEs)	6.5	25	Equation (7)	1.67 × 10^−6^	544.84	0.0075	0.969	This study
*Cladosporium cladosporioides* (LB)	2.0	25	Equation (4)	7.50 × 10^−5^	28.90	–	0.991	[70]
*Aspergillus niger* (SS)	2.0	100	Equation (4)	5.76 × 10^−4^	56.15	–	0.994	[71]
			Equation (5)	–	47.33	0.5416	0.999	
*Aspergillus niger* (DB)	2.0	27	Equation (4)	3.38	6.45	–	0.998	[67]
			Equation (5)	–	71.9	0.031	0.999	
*Lentinus sajor-caju* (CS)	2.0	30	Equation (4)	3.39 × 10^−2^	20.80	–	0.994	[72]
			Equation (5)	–	23.32	0.0133	0.993	
*Ustilago maydis* (DB)	5.5–6.5	25	Equation (4)	1.37 × 10^−2^	1.95	–	ns ^b^	[73]
			Equation (5)	–	17.16	0.0090	0.965	
*Mucor hiemalis* (DB)	2.0	100	Equation (4)	5.5 × 10^−1^	30.5	–	0.993	[74]
			Equation (5)	–	47.4	0.0307	0.999	
*Ganoderma applanatum* (DB)	2.0	25	Equation (4)	7.4 × 10^−1^	16.13	–	0.999	[75]
			Equation (5)	–	200	0.002	0.999	
*Rhizopus* sp. (DB)	2.0	25	Equation (4)	1.13 × 10^−2^	5.4509	–	0.986	[65]
			Equation (5)	–	8.0589	0.7730	0.841	

^a^ fungal species and pre-treatment of the biomass: living cell suspension (CS); melanin extracts (MEs); lyophilized biomass (LB); spore suspension (SS); dead biomass (DB). ^b^ not determined/not shown.

The results from this study are consistent with the hypothesis that fungal melanin plays an active role in the biosorption of toxic HMM, such as As(V) and Cr(VI). Melanin is an amorphous polymeric structure that concentrates in the fungal cell wall and offers a great number of heterogeneous binding sites to the sorption of metal ions [28]. Melanin is also a conductive material that mediates electron transport between fungal biomass and the solute, which might contribute to chemisorption by sharing or exchanging electrons between the sorbent and sorbate [76]. A previous study on the binding of copper by melanin extracts, intact cells, and albino mutants of the BF *Cladosporium resinae* and *Aureobasidium pullulans* demonstrated that the metal uptake capacity was higher in melanin extracts, followed by intact cells and, finally, in albino cells [77].

However, other studies have disputed the role of melanin as a metal biosorbent. An early study found no significant differences in the binding of copper by cultures of the melanized plant pathogen *Gaeumannomyces graminis*, either grown at low concentrations of this metal or additionally supplemented with tricyclazole, an inhibitor of DHN-melanin [78]. More recently, it has been proposed that melanin may have a role in binding metals and protecting fungi from toxic metals, but the main mechanisms might not necessarily be related to sorption processes [79,80]. Potisek [80] used different strains of dark septate endophytes of *Cadophora* spp., with different melanin contents, to investigate their tolerance to cadmium. The melanin content was positively correlated with a higher cadmium tolerance, but the accumulation of this metal was not. More contentious studies on *Exophiala pisciphila* even claimed that the inhibition of melanization in this fungus did not cause any remarkable effect on the tolerance of metal ions [43].

Indeed, the high inter- and intraspecific variability in metallotolerance observed among the studied black fungi (Table 2) suggest that there must be other physiologic mechanisms to cope with the toxic effects of toxic metals and metalloids, rather than relying solely on melanization. In this regard, the comparatively high biosorption capacity of BF to HMMs, in combination with low sorption rates, could be the result of an evolved strategy to, firstly, minimize the penetration of toxic metals into the cell and, secondly, sequester those that would eventually leak into the cytoplasm.

## 4. Conclusions

Several published studies have revealed that fungal biomass has good performance as a biosorbent of heavy metals and metalloids, in comparison to commercial materials such as ion exchange resins, activated carbon, and metal oxides. Fungal biosorption depends on parameters such as the used fungi and metal species but also the pH, temperature, biomass pre-treatments, and the presence of various ligands in solution. The fungal cell wall fraction seems to play an important role in the sorption of heavy metals, though the biosorption mechanisms are understood only to a limited extent. The ability of fungi to tolerate and biosorb specific heavy metals and metalloids has previously been evaluated and reviewed, as is the case for As(V) and Cr(VI), but the literature on the metallotolerance of extremophilic black fungi is still comparatively scarce.

To our knowledge, the present work provides the first in-depth survey of the metallotolerance of a strain collection of BF isolated previously from polluted sites. The obtained results indicate that there is broad inter- and intraspecific variability in metallotolerance, using As(V) and Cr(VI) as model HMMs. Living cultures of two hyper-metallotolerant strains of *Rhinocladiella similis* and *Exophiala mesophila* displayed a significant biosorption capacity but, conversely, sorption rates were comparatively slow in relation to other non-black fungi. This phenomenon could be explained by the interplay of two protective processes: an outer cell wall barrier and the inner cell wall biosorption of leaked metals and metalloids by melanin. These findings might contribute to the development of future strategies for the bioremediation of HMM pollution and need to be further investigated. Black fungi could also be valuable bioindicators of HMMs in natural and anthropogenic environments.

## Figures and Tables

**Figure 1 jof-10-00047-f001:**
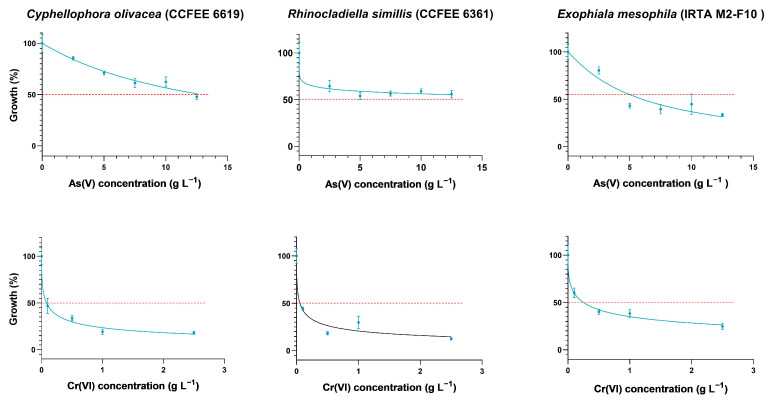
Effect of the concentration of As(V) (**top**) and Cr(VI) (**bottom**) on the average and standard deviation (solid bars, *n* = 6) of the tolerance index (TI) of selected metallotolerant fungal strains, measured from the radial growth in agar cultures after 30 days of incubation. Solid lines correspond to a fitted logistic model for the determination of the IC_50_ value.

**Figure 2 jof-10-00047-f002:**
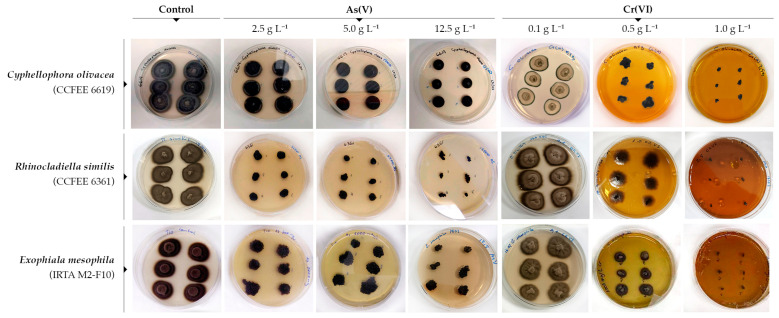
Macromorphological effects of the exposure to increasing concentrations of As(V) and Cr(VI) on fungal agar cultures of the metallotolerant black fungi.

**Figure 3 jof-10-00047-f003:**
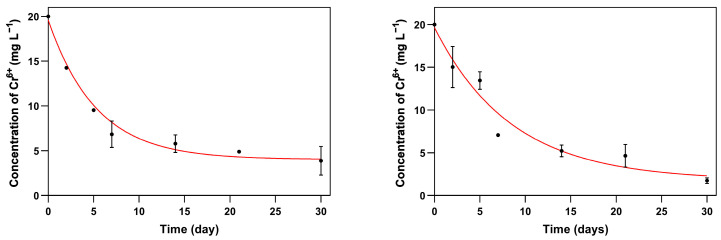
Fits of experimental data of Cr(VI) biosorption on living cultures (pH 6.5) of *Rhinocladiella similis* (**left**) and *Exophiala mesophila* (**right**) to the second-order kinetics and the Langmuir isotherm model (differential Equation (7)). Experimental data correspond to the average and standard deviation of three replicates.

**Figure 4 jof-10-00047-f004:**
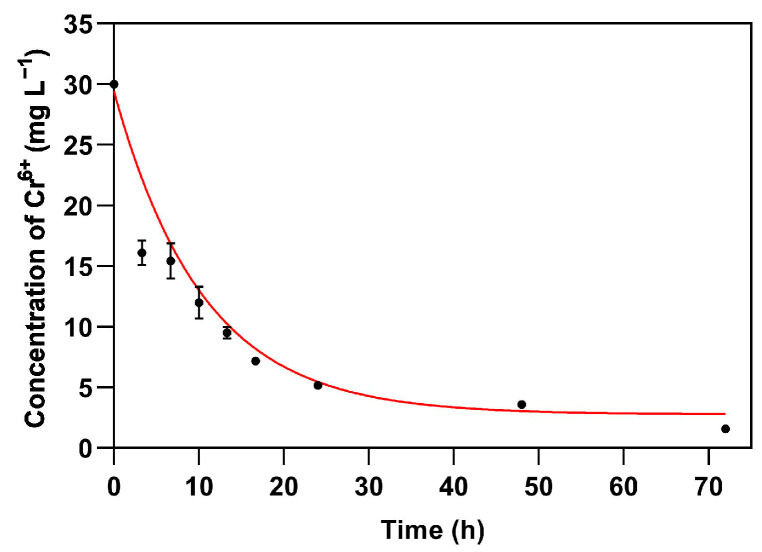
Fit of experimental data of Cr(VI) biosorption on melanin extracts of *Exophiala mesophila* to the second-order kinetics and the Langmuir isotherm model (differential Equation (7)). Experimental data correspond to the average and standard deviation of three replicates.

**Table 1 jof-10-00047-t001:** List of melanized fungal strains used in this study.

Species	Phylogenetic Group	Isolation Source	Strain nr ^a^	Accession nr ^b^
*Aulographina pinorum*	Asterinales	Diesel pump	CCFEE 6222	OR660094
*A. pinorum*	Asterinales	Diesel pump	CCFEE 6230	MZ573423
*Aureobasidium melanogenum*	Dothideales	Diesel car tank	CCFEE 6213	OR660095
*Au. melanogenum*	Dothideales	Diesel car tank	CCFEE 6234	OR660096
*Au. pullulans*	Dothideales	Gasoline car tank	CCFEE 5876	JX681059
*Au. pullulans*	Dothideales	Gasoline pump	CCFEE 6244	OR660097
*Cladosporium herbarum* ^c^	Cladosporiales	Diesel pump	CCFEE 6193	MZ573426
*Cl. herbarum*	Cladosporiales	Diesel pump	CCFEE 6192	OR660098
*Coniosporium uncinatum*	Dothideomycetes i.s. ^d^	Gasoline car tank	CCFEE 5820	JX681057
*Co. uncinatum*	Dothideomycetes i.s.	Gasoline pump	CCFEE 6149	MZ573424
*Cyphellophora olivacea*	Chaetothyriales	Biocide-treated monument	CCFEE 6619	MT472271
*Exophiala angulospora*	Chaetothyriales	Biocide-treated monument	CCFEE 6620	MT472272
*E. crusticola*	Chaetothyriales	Gasoline pump	CCFEE 6188	OR660099
*E. equina*	Chaetothyriales	Washing machine soap dispenser	CCFEE 5883	JX681045
*E. heteromorpha*	Chaetothyriales	Gasoline pump	CCFEE 6240	MZ573439
*E. heteromorpha*	Chaetothyriales	Diesel car tank	CCFEE 6150	OR660100
*E. lecanii-corni*	Chaetothyriales	Washing machine soap dispenser	CCFEE 5688	OR660101
*E. mesophila*	Chaetothyriales	Washing machine soap dispenser	CCFEE 5690	JX681043
*E. mesophila*	Chaetothyriales	Glued ceramics	IRTA M2-F10	OR660102
*E. oligosperma*	Chaetothyriales	Human patient	CBS 725.88	AY163551
*E. oligosperma*	Chaetothyriales	Diesel car tank	CCFEE 6139	MZ573441
*E. phaeomuriformis*	Chaetothyriales	Diesel car tank	CCFEE 6242	MZ573445
*E. xenobiotica*	Chaetothyriales	Gasoline car tank	CCFEE 5784	OR660103
*E. xenobiotica*	Chaetothyriales	Bathroom wet cell	CCFEE 5985	JX681024
*E. xenobiotica*	Chaetothyriales	Diesel car tank	CCFEE 6143	OR660104
*E. xenobiotica*	Chaetothyriales	Gasoline pump	CCFEE 6142	OR660105
*Knufia epidermis*	Chaetothyriales	Diesel car tank	CCFEE 6138	MZ573455
*K. epidermis*	Chaetothyriales	Gasoline car tank	CCFEE 5813	JX681055
*K. epidermis*	Chaetothyriales	Diesel car tank	CCFEE 6198	OR660106
*K. epidermis*	Chaetothyriales	Diesel car tank	CCFEE 6366	OR660107
*Neohortaea acidophila*	Mycospharellales	Lignite	CBS 113389	OL739260
*Rhizosphaera kalkholffii*	Dothideales	Diesel car tank	CCFEE 6144	OR660108
*Rhinocladiella similis*	Chaetothyriales	Diesel car tank	CCFEE 6361	MZ573467
*Scolecobasidium* cft *globale*	Venturiales	Diesel car tank	CCFEE 6363	MZ573464

^a^ CCFEE: Culture Collection of Fungi from Extreme Environments, Tuscia University; CBS: fungal collection of the Westerdijk Fungal Biodiversity Institute; IRTA: microbial collection of the Laboratory of Environmental Microbiology, Institute of Agrifood Research and Technology. ^b^ ITS1-5.8S-ITS2 ribosomal DNA sequences deposited in GenBank. ^c^ the strain reported as *Cladosporium herbarum* should be read as belonging to the corresponding species complex. ^d^
*Incertae sedis* (uncertain taxonomic placement).

**Table 2 jof-10-00047-t002:** Fungal TI and IC_50_ to As(V) and Cr(VI) after the incubation of agar cultures exposed to the highest and lowest tested concentrations during 30 days. IC_50_ was estimated by fitting the TI data to a logistic model (*r*^2^ > 0.9 unless stated otherwise). The strains from Table 1 that are not shown did not display any growth on the tested As(V) and Cr(VI) concentrations.

Species	Strain Nr	TI to As(V)		IC_50_ to As(V)	TI to Cr(VI)		IC_50_ to Cr(VI)
		(2.5 g L^−1^)	(12.5 g L^−1^)	(g As^5+^ L^−1^)	(0.1 g L^−1^)	(2.5 g L^−1^)	(g Cr^6+^ L^−1^)
*A. pinorum*	CCFEE 6222	0.61	– ^a^	3.87 (3.06–4.58)	–	–	–
	CCFEE 6230	0.44	–	1.70 (0.67–2.56)	–	–	–
*Au. melanogenum*	CCFEE 6213	0.49	–	2.42 (2.15–2.68)	–	–	–
*Au. melanogenum*	CCFEE 6234	0.66	0.31	4.14 (3.39–4.83)	–	–	–
*Au. pullulans*	CCFEE 6244	0.28	–	0.96 (0.69–1.21)	0.44	–	0.07 (0.05–0.09)
*Cl. herbarum*	CCFEE 6193	0.56	–	2.61 (2.27–2.90)	–	–	–
*Cl. herbarum*	CCFEE 6192	–	–	–	0.24	–	0.00 (0.0002–0.008)
*C. uncinatum*	CCFEE 5820	0.36	0.22	0.75 (0.28–1.26)	0.37	–	0.04 (0.02–0.06)
*C. olivacea*	CCFEE 6619	0.86	0.48	12.94 (11.53–15.09)	0.47	0.19	0.08 (0.05–0.12)
*E. crusticola*	CCFEE 6188	1.00	0.50	10.04 (8.94–11.61) ^b^	0.73	0.32	0.56 (0.36–0.84) ^b^
*E. equina*	CCFEE 5883	0.15	–	0.03 (0.0003–0.15)	0.43	–	0.04 ^c^
*E. lecanii-corni*	CCFEE 5688	0.32	–	0.20 (0.001–0.71)	0.65	0.46	1.01 (0.82–1.30) ^c^
*E. mesophila*	CCFEE 5690	0.42	–	1.50 (0.96–1.98)	–	–	–
	IRTA M2-F10	0.81	0.34	6.03 (5.19–6.94) ^b^	0.44	0.30	0.05 (0.02–0.09)
*E. oligosperma*	CBS 725.88	0.49	–	2.49 (1.62–3.20)	–	–	–
	CCFEE 6139	0.19	–	0.16 (0.02–0.44)	–	–	–
*E. phaeomuriformis*	CCFEE 6242	0.31	–	0.70 (0.32–1.09)	0.32	–	0.01 (0.003–0.03)
*E. xenobiotica*	CCFEE 5784	0.32	–	0.67 (0.15–1.28)	–	–	–
	CCFEE 5985	0.50	0.24	2.92 (1.59–3.96) ^b^	–	–	–
	CCFEE 6142	0.33	–	0.39 (0.04–0.93)	–	–	–
	CCFEE 6143	0.38	–	0.67 (0.16–1.25)	–	–	–
*K. epidermis*	CCFEE 5813	0.50	0.24	2.92 (1.59–3.96) ^b^	–	–	–
	CCFEE 6138	0.14	–	0.01 (~–0.15)	–	–	–
	CCFEE 6198	0.42	–	1.93 (1.54–2.27)	–	–	–
	CCFEE 6366	0.23	–	0.25 (0.06–0.53)	–	–	–
*N. acidophila*	CBS 113389	0.41	0.52	~	–	–	–
*R. similis*	CCFEE 6361	0.65	0.56	48.26 (16.14–~) ^b^	0.60	0.40	0.25 (0.18–0.33)
*Scolecobasidium*	CCFEE 6363	0.26	0.15	~	–	–	–

^a^: no significant growth; ^b^ 0.80 < *r*^2^ < 0.90; ^c^
*r*^2^ ≤ 0.80.

**Table 3 jof-10-00047-t003:** Fungal HMM biosorption capacity and removal efficiency. Incubations lasted 30 days, and results are expressed as the average and standard deviation of three independent experiments. One-way ANOVA comparisons were performed on the specific removal capacity for every metal, and non-significant differences are indexed (*n* = 3; *p* < 0.05). Significance letters for the Tukey test were added to the specific removal capacity data.

Fungus (Strain)	Type of Biomass	pH	Parameter (Units)	As(V)	Cr(VI)
*R. similis*	Living culture	6.5	Biomass (mg DM)	48.07 ± 6.05	43.33 ± 2.95
			Final concentration in solution (mg L^−1^)	18.30 ± 0.31	3.48 ± 1.33
			Specific removal capacity (mg g DM^−1^)	1.07 ± 0.32 ^AC^	11.50 ± 1.56 ^A^
			Removal efficiency (%)	8.5	82.6
		4.0	Biomass (mg DM)	45.28 ± 5.08	49.67 ± 0.01
			Final concentration in solution (mg L^−1^)	19.53 ± 0.31	<0.01
			Specific removal capacity (mg g DM^−1^)	0.33 ± 0.24 ^B^	≥12.79
			Removal efficiency (%)	2.3	100
*E. mesophila*	Living culture	6.5	Biomass (mg DM)	44.33 ± 1.91	46.90 ± 1.23
			Final concentration in solution (mg L^−1^)	18.04 ± 0.57	1.16 ± 0.25
			Specific removal capacity (mg g DM^−1^)	1.34 ± 0.42 ^A^	12.57 ± 3.00 ^A^
			Removal efficiency (%)	9.8	94.2
		4.0	Biomass (mg DM)	46.57 ± 6.12	53.57±1.55
			Final concentration in solution (mg L^−1^)	19.67 ± 0.21	<0.01
			Specific removal capacity (mg g DM^−1^)	0.21 ± 0.12 ^B^	11.20
			Removal efficiency (%)	1.67	100
	Dead cells	6.5	Biomass (mg DM) (mg L^−1^)	42.03 ± 0.63	46.40 ± 0.46
			Final concentration in solution	19.41 ± 0.25	5.16 ± 0.66
			Specific removal capacity (mg g DM^−1^)	0.41 ± 0.17 ^BC^	9.59 ± 0.47 ^A^
			Removal efficiency (%)	2.92	74.2

## Data Availability

All data of this research are reported in the main text and Appendix A.

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
