# Peer review of "The Metallotolerance and Biosorption of As(V) and Cr(VI) by Black Fungi"

_jof, 2024, doi:10.3390/jof10010047_

Round 1

Reviewer 1 Report

Comments and Suggestions for Authors

Dear authors,

Here are my suggestions to improve the quality of your work:

-        Introduce sources of arsenic and chromium contamination in urban areas to contextualize the choice of these elements in the study.

-        Line 131 – reference should be replaced to a published one; or the statement ends in [29,30].

-        Please provide the reason for growth in rich-agar PDA for 60 days and some cases have not verified growth (diameter measuring, line 110) and use of oligotrophic medium in order to “minimize growth” in section 2.5. As you worked for 30 days with the most promising fungi, why reduce growth? What you intended to simulate?

-        Information of time in lines 110 and 224 are not congruent.

-        In tolerance index test please provide a reference that interpret the isolates according high, moderate and low tolerance

-        Line 275 – please italicize the name Aspergillus and check for the others. In my opinion, you should suppress the expression genera in all cases because two reasons: 1. phyla and order in the introduction were not announced; 2. Considering a journal on fungal life, it is welcome to write (gen.) or spp.   

-        Line 286, please italicize in vitro

-        Line 291, same for Exophiala

-        Lines 322-323, please provide the mechanism of that affirmation.

-        It recommended to suppress lines 476-478 as long as it was available for further access and reading.

Author Response

For research article

Response to Reviewer 1 Comments

1. Summary

Point-by-point response to Comments and Suggestions for Authors

Comments 1: Introduce sources of arsenic and chromium contamination in urban areas to contextualize the choice of these elements in the study.

Response 1:

This issue has been addressed by adding the following sentences in the introduction section:

“Arsenic exposure to humans can be attributed to various sources, including natural deposits, agricultural pesticides, and industrial effluents.” Page 1, line 44-45

“The majority of chromium releases into the environment originate from industrial activities, particularly from the processing and manufacturing of chemicals, minerals, steel, leather tanning, textile dyeing, metallurgical operations, and various other industrial processes.” Page 2, line 50-54

Comments 2: Line 131 – reference should be replaced to a published one; or the statement ends in [29,30].

Response 2: The unpublished reference Medina-Armijo et al (in preparation) has been removed.

Comments 3: Please provide the reason for growth in rich-agar PDA for 60 days and some cases have not verified growth (diameter measuring, line 110) and use of oligotrophic medium in order to “minimize growth” in section 2.5. As you worked for 30 days with the most promising fungi, why reduce growth? What you intended to simulate?

Response 3: Because most BF strains grow slowly compared to several other fungi strains, the study was conducted up to 60 days in order to account for the potential long-term adaptation. A couple of additional sentences have been added to compare and discuss about the effect of the incubation time on metallotolerance, which seems to be indeed relevant for the selected strains.

PDA has been used as a standard mycological medium in metallotolerant assays, which is suitable for the cultivation of most ascomycetes. The definition of the medium as “oligotrophic” refers only to the liquid broth in which pre-grown fungal biomass was incubated to perform the biosorption kinetic experiments.

 Comments 4: Information of time in lines 110 and 224 are not congruent.

Response 4: The information presented in these sentences corresponds to an intermediate incubation period of 30 days. The results from other tested incubation times are presented in Table S1. To avoid confusion, the following sentences have been changed:

“Each BF strain was inoculated six times with an inoculation loop on each agar plate containing a defined concentration of As(V) or Cr(VI). The radial growth of each fungal colony was measured by averaging orthogonal diameters, and this value was then averaged again for all six colonies from every single plate. These measurements were repeated after 7, 15, 30, 45, and 60 days of incubation, which was performed at 25°C under dark conditions. This prolonged timeframe was stablished to account for the potential long-term adaptation of slow growing BF strains. Unamended control plates were also included.” Page 4, line 115-122

The effect of higher and lower metal exposition times has also briefly been discussed in the main text.

Comments 5: In tolerance index test please provide a reference that interpret the isolates according high, moderate and low tolerance

Response 5: Given the diversity of fungi and methods, there si no specific bibliographic reference that provides detailed information on how to classify fungal metallotolerance in a standardized way. Our results have been interpreted in relative terms, on the basis of comparing the TI and IC50 values determined for the different strains of the tested collection. When possible, comparisons with similar previously published studies have also been performed.

Comments 6: Line 275 – please italicize the name Aspergillus and check for the others. In my opinion, you should suppress the expression genera in all cases because two reasons: 1. phyla and order in the introduction were not announced; 2. Considering a journal on fungal life, it is welcome to write (gen.) or spp.

Response 6: This term has been italianized. Fungal strains identified only at the genus level have been written as spp.

Comments 7: Line 286, please italicize in vitro

Response 7: This term has been italianized.

Comments 8: Line 291, same for Exophiala

Response 8: This term has been italianized.

Comments 9: Lines 322-323, please provide the mechanism of that affirmation.

Response 9: An explanation about different possible mechanisms has been added.

Comments 10: It recommended to suppress lines 476-478 as long as it was available for further access and reading.

Response 10: This sentence has now been suppressed

Reviewer 2 Report

Comments and Suggestions for Authors

The manuscript entitled “Metallotolerance and Biosorption of Toxic Metals and 2

Metalloids by Black Fungi » deals with the tolerance to toxic concentrations of As(V) and Cr(VI) anions of a collection of 34 melanized fungi isolated previously from anthropogenic contaminated sites.

 The manuscript is well written, structured and informative.

It must be improved before publication: major revisions.

Below is a list of requested corrections:

The title is too general, needs to me more precise as the authors deal with especially As and Cr

Line 20: In the abstract, define the acronym DM before use it

Line 59: a reference is not filled in

Line 109: ‘Each BF strain was six-point-seeded with an inoculation loop in every agar plate” unclear - understandable only seeing the photos

Table 1: To which phylogenetic groups do Asterinales and Venturiales belong ?

Line 168: define the acronym TI before use it

Line 204: paragraph 3.1. Metallotolerance of Black Fungi – In this part, numerous errors or incoherence in the text with table 2

Line 206-208 “Of those, 17 (50%) displayed measurable growth on agar cultures exposed to As(V) at the maximum tested concentration of 12.5 g- As5+ L-1”. Ten strains were found in the table 1 instead 17?

Which values of TI or of CI 50 were used to identify the first group described in the text? “The first group of tolerant strains primarily belonged to the Chaetothyriales (Cyphellophora olivacea, Exophiala crusticola, E. equina, E. lecanii-corni, E. mesophila,E. oligosperma, E. phaeomuriformis, E. xenobiotica, and Rhinocladiella similis)”

Why some non-tolerant strains do not appear in Table 2 (E. angulospora) and others do (E. heteromorpha, Scolecobasidium, R. kalkholffii)?

In general, the presentation of Table 2 is difficult to understand.

Idem E. crusticola with a high growth with As 2.5 g/L is not discussed in the text

Line 242

“Conversely, the tolerance of E. crusticola CCFEE 6188 to Cr(VI) was lower when compared to other fungi, as it had TI of 0.47 at 0.1 g-Cr6+ L-1 after 30 days of exposure but it grew scarcely at the higher tested concentrations (its IC50 was 0.56 g-Cr6+ L-1).”

The value in the Table 2 is 0.73 instead of 0.47

Line 246: E. mesophila P6-8-M2 not cited in the Tables 1 and 2. Why?

Line 259: List strains in alphabetical order for easier reading

Figure 1: the y-axis legend is not % inhibition and does not correspond to the figure title

This figure needs to be corrected.

Line 309: C. uncinatum write it in italics

Line 312: IC50 values of 6 – 8 g-As5+ L-1, do not fit with the Table 2

Line 338: “meristematic growth”? Explain this term for fungi?

Figure 2: The text appears blurred - quality to be improved

Table 3: The results of ph values on bisorption are not given for R. similis. Why?

Line 408: Are differences between pH 4 and pH 6.5 significant?

Line 464: Cladosporium cladosporioides written with a minuscule

Table 4: The interest of table 4 is to be qualified as many fungi are phylogenetically distant from the strains studied. If this table is retained, correct the references as they are out of sync.

Author Response

For research article

Response to Reviewer 2 Comments

1. Summary

Point-by-point response to Comments and Suggestions for Authors

Comments 1: The title is too general, needs to me more precise as the authors deal with especially As and Cr

Response 1: The title has been changed as “Metallotolerance and Biosorption of As(V) and Cr(VI) by Black Fungi.”

Comments 2: Line 20: In the abstract, define the acronym DM before use it

Response 2: The full term “dry matter” has been used in the abstract.

Comments 3: Line 59: a reference is not filled in

Response 3: The reference number (1,13) has been included.

Comments 4: Line 109: ‘Each BF strain was six-point-seeded with an inoculation loop in every agar plate” unclear - understandable only seeing the photos

Response 4:

Page 4, line 115-122: A more precise description for this method has been included:

“Each BF strain was inoculated six times with an inoculation loop on each agar plate containing a defined concentration of As(V) or Cr(VI). The radial growth of each fungal colony was measured by averaging orthogonal diameters, and this value was then averaged again for all six colonies from every single plate. These measurements were repeated after 7, 15, 30, 45, and 60 days of incubation, which was performed at 25°C under dark conditions. This prolonged timeframe was stablished to account for the potential long-term adaptation of slow growing BF strains. Unamended control plates were also included.”

Comments 5: Table 1: To which phylogenetic groups do Asterinales and Venturiales belong?

Response 5: These two orders belong to the class Dothideomycota (division Ascomycota). For the sake of simplicity, only the order classification is employed in this table.

Comments 6: Line 168: define the acronym TI before use it

Response 6:

Page 5, line 178

This acronym has now been defined.

Comments 7:

Line 286, please italicize in vitro

Response 7: This term has been rewritten in italics.

Comments 8: Line 291, same for Exophiala

Response 8: This name has been rewritten in italics.

Comments 9: Line 204: paragraph 3.1. Metallotolerance of Black Fungi – In this part, numerous errors or incoherence in the text with table 2

Line 206-208 “Of those, 17 (50%) displayed measurable growth on agar cultures exposed to As(V) at the maximum tested concentration of 12.5 g- As5+ L-1”. Ten strains were found in the table 1 instead 17?

Why some non-tolerant strains do not appear in Table 2 (E. angulospora) and others do (E. heteromorpha, Scolecobasidium, R. kalkholffii)?

In general, the presentation of Table 2 is difficult to understand.

Response 9: The reason is that Table 2 only presents the results for the strains that exhibited growth with any of the tested metal concentrations, while Table 1 provides a complete list of the tested strains. From the 34 strains from Table 1, there were 5 strains (R. kalkholffii, E. heteromorpha, Au. pullulans) that did not display any growth under metal exposure and were therefore not included in Table 2. This statement is provided in the legend of Table 2.

Strains not included in table 2:

Au. pullulans CCFEE 5876

Cladosporium herbarum CCFEE 6193

Cl. Herbarum CCFEE 6192

Coniosporium uncinatum CCFEE 6149

Exophiala angulospora CCFEE 6620

E. heteromorpha CCFEE 6240

E. heteromorpha CCFEE 6150

Rhizosphaera kalkholffii CCFEE 6144

Scolecobasidium cft globale CCFEE 6363

Comments 10: Which values of TI or of CI 50 were used to identify the first group described in the text? “The first group of tolerant strains primarily belonged to the Chaetothyriales (Cyphellophora olivaceaExophiala crusticolaE. equinaE. lecanii-corniE. mesophila,E. oligospermaE. phaeomuriformisE. xenobiotica, and Rhinocladiella similis)”

Response 10:

Page 6, line 244-246

Metallotolerant strains have been defined on grounds of an IC50 value greater than X. This criterion has now been included in the manuscript.

Comments 11: Idem E. crusticola with a high growth with As 2.5 g/L is not discussed in the text

Line 242

Response 11: the discussion of E. crusticola with a high growth with As 2.5 g/L is mentioned in page 6, line 250-258.

“A few chaetothyrialean strains stood out from the rest because of their remarkable level of metallotolerance, such as Exophiala crusticola CCFEE 6188. This particular strain had the highest tolerance to As(V), with average TI values that after 30 days ranged from 1.00 at 2.5 g-As5+ L- to 0.50 at 12.5 g-As5+ L-1. The corresponding modelled IC50 estimate was 10.0 g-As5+ L-1…”

Comments 12: Conversely, the tolerance of E. crusticola CCFEE 6188 to Cr(VI) was lower when compared to other fungi, as it had TI of 0.47 at 0.1 g-Cr6+ L-1 after 30 days of exposure but it grew scarcely at the higher tested concentrations (its IC50 was 0.56 g-Cr6+ L-1).”

The value in the Table 2 is 0.73 instead of 0.47

Response 12: The value has been corrected.

Comments 13: Line 246: E. mesophila P6-8-M2 not cited in the Tables 1 and 2. Why?

Response 13: We mistakenly took the sampling coding (P6-8-M2) instead of the collection reference (IRTA M2-F10) for this particular strain. The error has now been corrected.

Comments 14: Line 259: List strains in alphabetical order for easier reading

Response 14: The strains have been ordered alphabetically.

Comments 15: Figure 1: the y-axis legend is not % inhibition and does not correspond to the figure title. This figure needs to be corrected.

Response 15: The reviewer is right, the y-axis of Figure 1 actually corresponds to the percentage of growth in relation to the control, which is the complementary value of the percentage of inhibition. This error of the axis legend has now been amended.

Comments 16: Line 309: C. uncinatum write it in italics

Response 16: This term has been rewritten in italics.

Comments 17: IC50 values of 6 – 8 g-As5+ L-1, do not fit with the Table 2

Response 17: These values have been revised ​​and corrected.

Comments 18: Line 338: “meristematic growth”? Explain this term for fungi?

Response 18:

Add the following sentence to clarify an explanation of meristematic growth:

Page 10, line 368-375 Meristematic growth is defined as the development of slowly expanding and irregular colonies, sometimes defined as “cauliflower-like colonies”, that are the result of the isodiametric enlargement and subdividing of cells. This is an accepted morphological term in myology that commonly corresponds to an adaptative response to environmental stress. The sentence has modified in the manuscript in order to clarify this concept.

Comments 19: Figure 2: The text appears blurred - quality to be improved

Response 19: A higher resolution image has been included in the manuscript.

Comments 20: Table 3: The results of ph values on bisorption are not given for R. similis. Why?

Response 20:  Because of budgetary and logistic limitations, we decided to take E. mesophila as a model organism and we did not perform this experiment with R. similis.

Comments 21: Line 408: Are differences between pH 4 and pH 6.5 significant?

Response 22: In contrast to the As(V) studies, significant differences were observed between species and treatments in the Cr studies. To clarify this, we include the following sentence: page 13, line 427-428

Interestingly, fungal incubations at pH 4.0 significantly (p>0.05) enhanced the biosorption of Cr(VI), but not that of As(V), compared to analogous experiments at pH 6.5 (Table 3).

Comments 22: Line 464: Cladosporium cladosporioides written with a minuscule

Response 22: This species name has been written correctly

Comments 23: Table 4: The interest of table 4 is to be qualified as many fungi are phylogenetically distant from the strains studied. If this table is retained, correct the references as they are out of sync.

Response 23: We think that this table is indeed relevant for comparative purposes with other unrelated fungal species. The contained references have been verified and, when necessary, corrected.

Reviewer 3 Report

Comments and Suggestions for Authors

Abstract

Background: address in a broad way why it is important to know about the metal removal capacity of these fungi.

Introduction

Page 2 line 59

Include the bibliographic reference that was not included.

Materials and methods

Explain why in the metallotolerance assays, the fungi were incubated for 60 days.

Results and discussion

3.1.     Metallotolerance of Black Fungi  

Although it is indicated to consult Table S1, it is necessary to describe in a simplified and clear manner the most relevant results at 7, 15, 45, and 60 days of incubation.

Page 8 lines 268-270

Explain in greater detail why Chromium is more toxic to fungi than arsenic.

Page 8 line 275

Italicize the genus Aspergillus.

Page 9 line 309

Italicize the C. uncinatum

References

Put italics in the scientific names of bibliographic reference number 32.

Put italics in the title of the journal of bibliographic reference number 46.

Author Response

For research article

Response to Reviewer 3 Comments

1. Summary

Point-by-point response to Comments and Suggestions for Authors

Comments 1: Introduction  Page 2 line 59

Include the bibliographic reference that was not included.

Response 1: These bibliographic reference numbers have been included (1,13).

Comments 2: Line 168: define the acronym TI before use

Response 2: The acronym for TI has been defined.

Comments 3: Results and discussion

Metallotolerance of Black Fungi 

Although it is indicated to consult Table S1, it is necessary to describe in a simplified and clear manner the most relevant results at 7, 15, 45, and 60 days of incubation.

Page 8 lines 268-270

Response 3: Table S1 shows the raw results for all incubation times and strains. We show the results of the 30th day of incubation in the manuscript because it is the most representative of the long-term effects of exposure to the metals. However, we have included a couple of sentences on the results with the other tested incubation times.

Comments 4: Explain in greater detail why Chromium is more toxic to fungi than arsenic.

Page 8 line 275

Response 4: In general, Cr(VI) is more toxic than As(V) to fungi because it is a more potent oxidizing agent and can form complexes with the fungus's essential metals. It can also oxi-dize proteins, which can disrupt enzymatic functions [3,8]

Page8, line 299-301

Comments 5: Italicize the genus Aspergillus. Page 9 line 309; Italicize the C. uncinatum

Response 5: This name has been rewritten in italics.

Comments 6: References

Put italics in the scientific names of bibliographic reference number 32.

Put italics in the title of the journal of bibliographic reference number 46.

Response 6: The scientific names and journal titles have been rewritten in italics

Round 2

Reviewer 2 Report

Comments and Suggestions for Authors

I thank the authors for their detailed answers to all remarks

Some small mistakes need to be corrected, please.

Table 2 contains 28 strains instead 29.

 I advise to respect the order of the strains, to make table 2 easier to read, as in Table 1: i.e. Coniosporium uncinatum after Cladosporium herbarum; If C. herbarum CCFEE 6192 is included, introduce its name. Scoleobasidium after R. similis.

Figure 1: strain names are slightly cropped. The number of C. olivacea is CCFEE 6619 and not 6919

Line 373 E. oligosperma CCFEE 6327 is not cited in Table 1

Author Response

For research article

Response to Reviewer 2 Comments

1. Summary

Point-by-point response to Comments and Suggestions for Authors

Comments 1: Table 2 contains 28 strains instead 29.

Response 1: The number has been changed as “28”

Comments 2: I advise to respect the order of the strains, to make table 2 easier to read, as in Table 1: i.e. Coniosporium uncinatum after Cladosporium herbarum; If C. herbarum CCFEE 6192 is included, introduce its name. Scoleobasidium after R. similis.

Response 2: order of strains changed as table 1

Comments 3: Figure 1: strain names are slightly cropped. The number of C. olivacea is CCFEE 6619 and not 6919

Response 3: Strain name improvement. Encoding 6619 has been updated

Comments 4: Line 373 E. oligosperma CCFEE 6327 is not cited in Table 1

Response 4: E. oligosperma CCFEE 6327, mentioned in the discussion (citation number 48), is not part of this study. It is referenced to illustrate the morphology under stress conditions. Line 376